# Drought Risk Evaluation in Iran by Using Geospatial Technologies

**Abdolreza Ansari Amoli [1,2,\*], Hossein Aghighi [3] and Ernesto Lopez-Baeza [2]**

1    Remote Sensing & GIS Department, Iranian Space Agency, No. 34, Sayeh St.,
     Vali Asr Ave., Tehran 1967734114, Iran
2    Environmental Remote Sensing Group (Climatology from Satellites), Earth Physics & Thermodynamics
     Department, Faculty of Physics, University of Valencia, 46100 Valencia, Spain; ernesto.lopez@uv.es
3    Center for Remote Sensing and GIS Research, Faculty of Earth Sciences, Shahid Beheshti University,
     Tehran 1983969411, Iran; h_aghighi@sbu.ac.ir
\*    Correspondence: anab@uv.es

**Abstract:** A drought risk map has been developed at the national scale by using remote-sensing satellite data over Iran by combining output layers resulting from three main components of a risk-evaluation procedure including *Hazard Quantification* (HQ), *Vulnerability Assessment* (VA) and *Identification of Elements at Risk* (IER) in a GIS environment. In this respect, *Drought Severity* (DS) was calculated by using the monthly *Normalized Difference Vegetation Index* (NDVI) (over 31 years from 1986–2016). Iran landcover classification and a slope map, population density maps, and irrigated farm percentages at the provincial scale were utilized within the drought risk evaluation (DRE) process. The final risk map reveals that the northwest of the country, with a climate similar to the central European weather conditions, is exposed to the maximum drought risk. In contrast, the areas with an arid climate, mainly located in the middle of Iran, exhibits minimum risk against drought. Based on the risk map, the southern part of the Caspian Sea shows very low drought risk due to the moderate and subtropical climate in this region. The outputs of this research will provide advice and warnings to help decision makers reduce drought risk consequences after prioritizing risk areas at the administrative scale.

**Keywords:** drought risk maps; exposure to drought; hazard quantification; remote sensing; vulnerability to drought

## 1. Introduction

Drought, which sometimes is described as a temporary climatic event due to lack of rain, is one of the most complicated phenomena to affect people's lives [1]. Based on the *World Disaster Report*, drought and famine, which have caused at least 275,000 deaths since 1994, can be regarded as the most fatal hazards of recent decades. This represents almost 50 percent of the total deaths caused by all types of natural hazards [2].

Drought follows a slow and lifelong pattern of influencing various features of the economy [3], society [4,5], and the environment [6]. In fact, the consequences of drought often endure for months and even years after the drought has ended. Moreover, drought damages are not structural and disseminate over larger geographical areas than other natural disasters. The nonstructural specification of drought consequences has certainly prevented the development of accurate, reliable, and timely estimations of drought's severity. Consequently, management of drought impact is more complex than that of the other types of hazards [6].

Iran, as a country located in a drought-prone area, has been influenced by frequent drought events during the last three decades with a particularly severe drought occurring from 1999 to 2002. This drought's damages were $3.5 billion, including losses in agricultural crops, livestock, and other items. This finally resulted in a strong reduction of agricultural

crops and products. Since that time, most water resources in the country have dried up severely [6].

The consequences mentioned above are mostly related to the traditional reactive method of drought management called "crisis management", which deals with coping with the direct effects of drought. As an essential policy, all drought-prone regions should improve their national drought plans and strategies by applying a change of paradigm from crisis management to risk management, insisting on developing activities and preparedness before the drought [7].

During recent decades, drought managers have distinguished the essentials of drought-risk management with the aim of mitigating drought impacts [8]. In most research on drought-risk assessment, the final risk map shows only drought hazard [9–13], vulnerability [14,15], exposure [16], or hazard and vulnerability [17]. By considering that Earth observation satellite techniques offer new opportunities to understand drought risks [18], a number of references, some of them mentioned above [5,7,8,13], have used remote-sensing data to analyze drought risk. The objective of this paper is the assessment (mapping) of drought risk in Iran by quantifying the three elements of risk namely hazard, vulnerability, and exposure, by using NOAA/AVHRR remote-sensing satellite data from 1986 to 2016, a land cover map, and a slope map, as well as statistical data such as population density and the percentage of irrigation farms to provide a national and provincial scale risk map in a GIS-based environment. To the authors' knowledge and review, this is the first work on drought risk in Iran in which the three main phases of risk management, i.e., hazard, vulnerability, and exposure, have been assessed by using satellite imagery and GIS, in a single research work.

The structure of the paper begins with the Materials and Methods section with a brief description of the study area in terms of geographical and climatological conditions as well as the datasets used in the research. An overview of the main components of disaster risk assessment and their conceptions will be described next. The discrepancies between two major components of a disaster risk assessment, namely risk analysis and risk evaluation, will also be elaborated upon in this section. Considering that the main aim of this research is drought-risk quantification, the focus will be on risk evaluation, which is the quantitative part of a risk-assessment procedure. In this regard, three major steps of risk evaluation including *Hazard Quantification* (HQ), *Vulnerability Assessment* (VA), and *Identification of Elements at Risk* (IER) will be discussed. Afterward, literature on methodologies, techniques, and indices used to quantify drought risk in Iran and at the international level will be reviewed. The methodology of the research is described, including a flow diagram. Finally, after explaining the results of the methodology, they will be discussed and conclusions will be given in detail.

## 2. Materials and Methods

### 2.1. Study Area

The study area is Iran, with a total surface of 1,648,195 km$^2$ located in the arid and semi-arid sphere of the world; the upper left and lower right geographical coordinates of Iran are shown in Figure 1. Figure 2 represents the aridity map of Iran obtained from 51 weather stations covering a 30-year statistical period (1981–2010), showing that a large part of Iran is covered by an arid and hyper-arid climate, respectively, covering approximately 26% and 54% of the country [19]. The temporal and spatial rainfall distributions are variable and non-uniform; although merely 10% of rainfall occurs during the hot and dry seasons in the central, southern, and eastern regions of the country [20]. The annual rainfall distribution indicates that 74% of the country (122.5 million ha) benefits from less than 200 mm rainfall (less than 1/3 of the global average) [21]. Nearly 52% of the annual rainfall and snowfall in only 25% of the country, which exposes some areas to drought; these areas are expected to face serious crises in the near future.

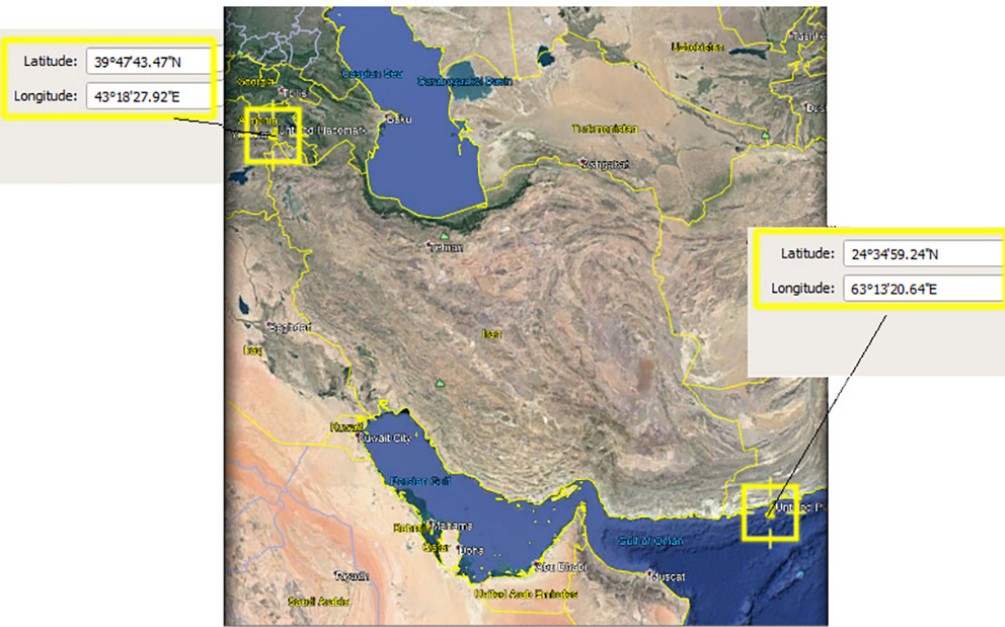

**Figure 1.** Iran upper left and lower right geographical coordinates (Google Earth).

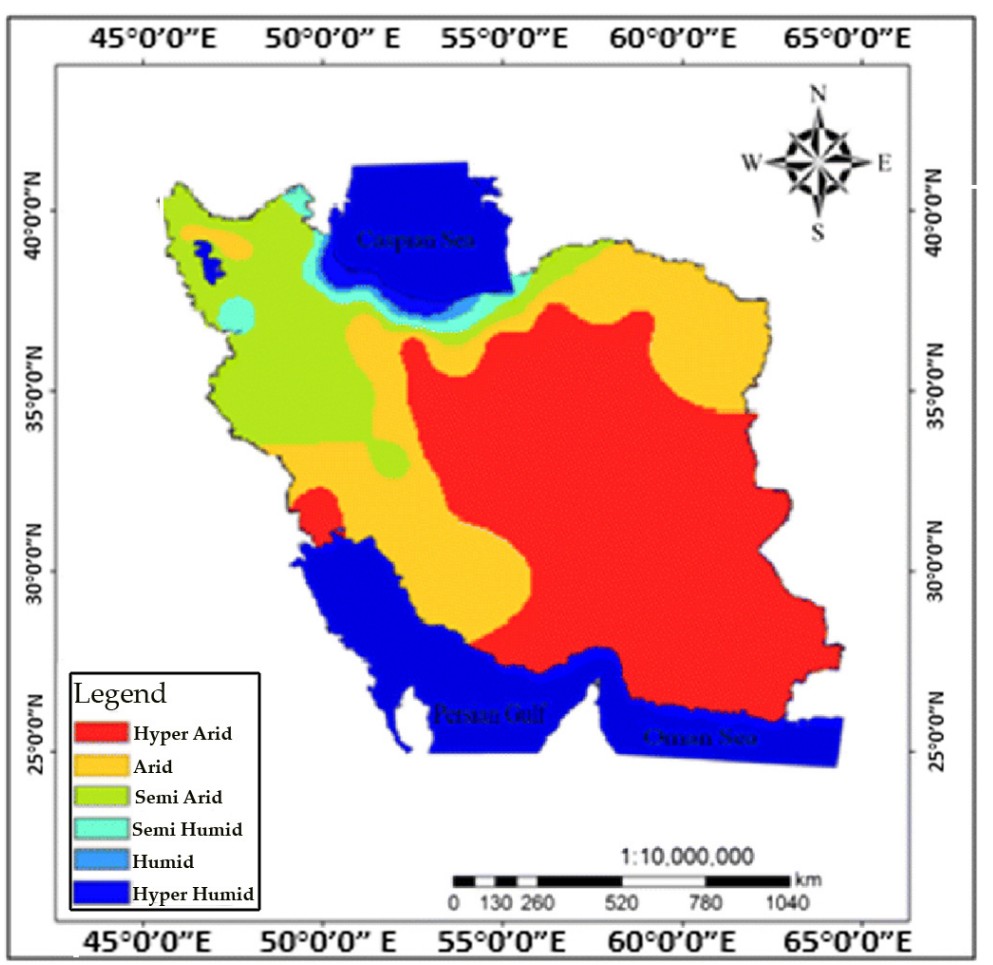

**Figure 2.** Iran aridity map (1981–2010).

### 2.2. Long-Term Air Temperature and Precipitation Trends in Iran

According to Intergovernmental Panel on Climate Change (IPCC) reports, the air temperature in the Middle East will increase by up to 2 °C, and there will be a decline in precipitation by 20% [22,23] between 2020 and 2040. Among Middle Eastern countries, Iran is very vulnerable to climate change with an increase of 2.6 °C in mean temperatures and a 35% decline in precipitation during the abovementioned period of time [24–27].

Figure 3 shows the temporal variations of mean surface air temperature and mean total surface precipitation from 1988–2018 over Iran for winter (DJF) and summer (JJA) periods. The figure shows that the air temperature increased significantly both in the minimum and maximum values of winter and summer periods. In contrast, the variable of surface precipitation declined dramatically in minimum and maximum records of summer and winter periods [28].

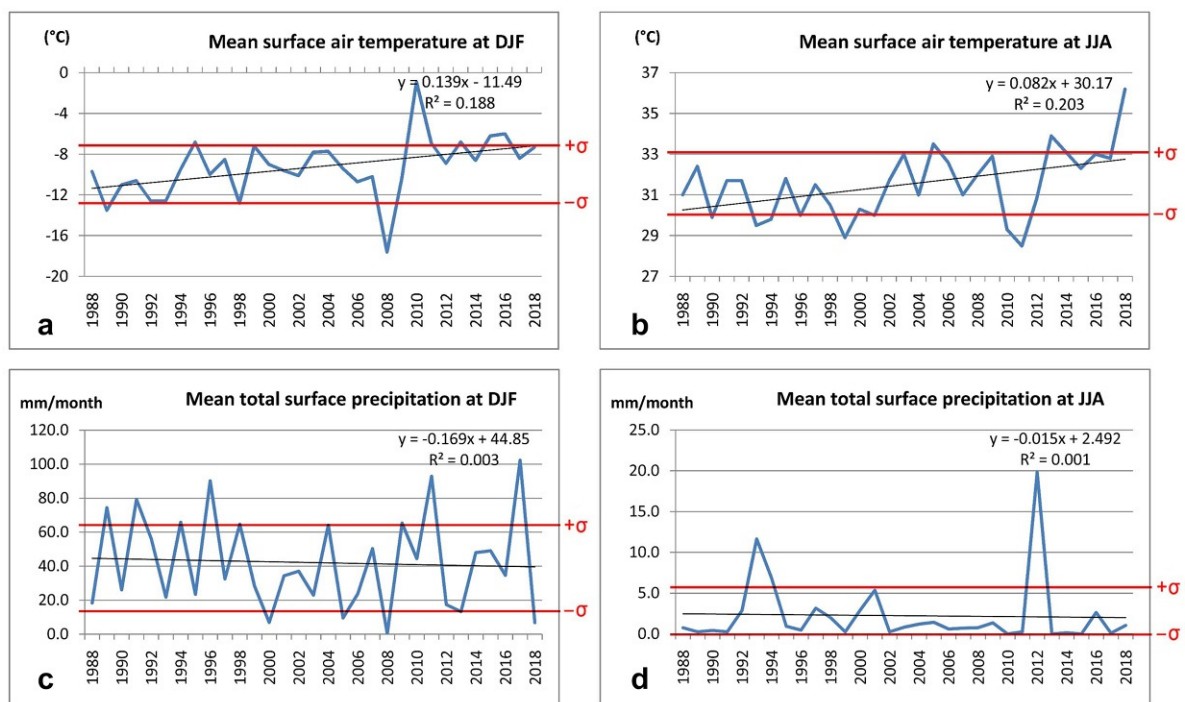

**Figure 3.** Temporal variations of Iran mean surface air temperature (blue lines) in winter (**a**) and summer (**b**), and Iran mean total surface precipitation (blue lines) in winter (**c**) and summer (**d**) from 1988–2018. The black lines show the trends for each variable during the period of study.

### 2.3. Disaster Risk Assessment

An important component of a disaster risk-management process is risk assessment [29]. The diagram shown in Figure 4 indicates the main components of a disaster risk-assessment system. As can be seen, the output of disaster risk assessment is a list of advice for disaster reduction which enables high-ranking managers and principle responsible organizations to make correct decisions about disaster mitigation and impact reduction. For example, a piece of advice based on the prioritization of disaster risk in different regions assists governors in making optimum decisions about accurate budget allocation to the disaster-stricken areas.

According to Figure 4, there are two general parts called "*Disaster Risk Analysis*" (qualitative phase) and "*Disaster Risk Evaluation*" (quantitative phase), which are described in the following paragraphs.

#### 2.3.1. Disaster Risk Analysis (Qualitative Phase)

In disaster risk analysis, there are two main steps to be undertaken, including the identification of causes and potential impacts. Each disaster is due to a number of reasons and causes, and proper identification of these causes helps researchers to predict possible

future disasters [30]. For example, precipitation is the most significant cause of drought. Basically, decreasing precipitation rates can often (although not necessarily always) be seen as an indicator of possible future drought. In other words, precipitation can be considered as an alert or precursor of drought in a region [31].

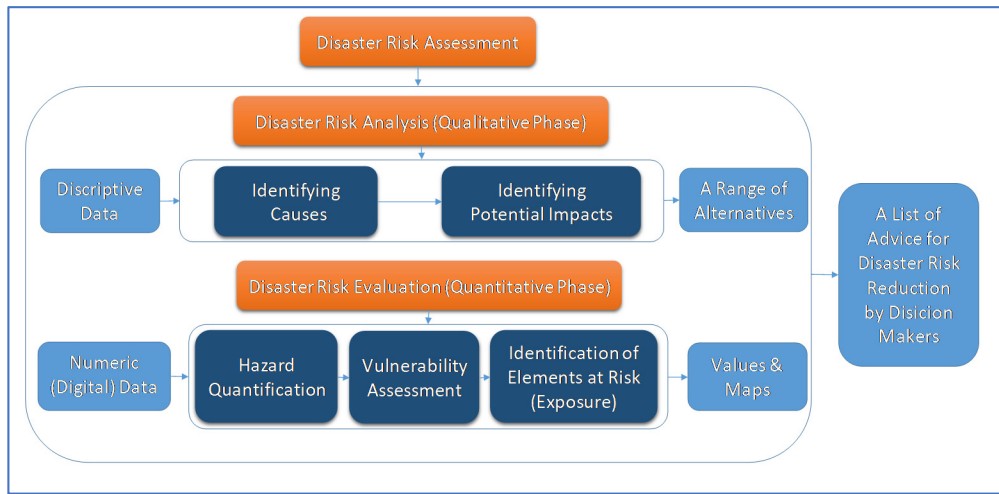

**Figure 4.** Disaster risk-assessment diagram.

The most important step of disaster risk analysis is determining the major prospective impacts of a disaster. Regarding droughts, one of the dominant impacts is agricultural crop loss, which results in losing money [32]. As shown in Figure 4, the inputs for disaster risk analysis are descriptive. For example, lack of precipitation can lead to a moisture deficit in the soil; a situation that, after a few months, will result in crop failure and financial loss. As a consequence, farmers will not be able to invest in the mechanisation of their farms and, eventually, the agricultural mechanisation industry will go bankrupt.

The outcome of the risk analysis is a list of alternatives to cope with the impacts of drought.

For example, one alternative could be the development of a pressurised or drip irrigation system to reduce water consumption in agricultural fields. Another solution is to plant water-resistant or drought-compatible crops and vegetation. The use of rainwater-harvesting systems for the optimal use of rainfall in drought conditions can be considered as another alternative to mitigate the impacts of drought.

2.3.2. Disaster Risk Evaluation

Disaster risk evaluation includes three main steps as follows:
- Hazard quantification (HQ);
- Vulnerability assessment (VA); and
- Identification of exposure; also called elements at risk (IER).

As shown in the diagram of Figure 4, the input data for disaster risk evaluation are numerical (digital) data, and outputs are values and maps. Specific techniques and technologies should be used in risk evaluation in order to work with quantified data.

In the following sections, three steps will be described for the quantitative phase.

- Hazard quantification (HQ):

The verbal meaning of "*hazard*" based on Black's Law Dictionary is "*A danger or risk lurking in a situation which by change or fortuity develops into an active agency of harm*" [33]. According to the description given by the Canadian Center for Occupational Health and Safety website [34], a *hazard* is "*Any source of potential damage, harm or adverse health effects on something or someone*". Based on the above definitions, all types of disasters can, in fact, be regarded as hazards. However, the mathematical concept of a *hazard* implies a probability

or likelihood of a hazardous event occurrence based on the number of times it happens in a specific period of time [35]. As a result, the general meaning of a hazard, as considered in this study, will be "*the frequency (probability) of occurrence of a drought event in a region with a specific intensity*". In order to calculate drought frequency, first of all, a parameter is needed to quantify the number of drought events in the past. This procedure is called "*Hazard Quantification*" (HQ). The most important parameter for HQ is *Drought Severity* (DS), which demonstrates the size or magnitude of drought. In this research, satellite-based vegetation indices are used to calculate the DS index.

The most commonly used remotely sensed vegetation index in drought studies, the *Normalized Difference Vegetation Index* (NDVI), is also applied to calculate drought severity. NDVI is a measure of the difference in reflectance between near-infrared (NIR) and red (R) wavelength ranges. Rouse et al. (1974) [36] introduced NDVI, computed as

$$(\text{NIR} - \text{R}) / (\text{NIR} + \text{R}). \tag{1}$$

The theory behind NDVI is based on the absorption by vegetation in the red and reflectance in the NIR. NDVI values range between $-1$ and $1$, with values greater than 0.5 indicating dense vegetation and values less than 0 demonstrating no vegetation [37].

The main statistical relation for severity index (SI) is:

$$\text{SI} = \frac{X - \overline{X}}{\sigma}, \tag{2}$$

where $X$ is the mean value of the variable under consideration in a specific period of time, $\overline{X}$ is the long-term mean of $X$ and $\sigma$ denotes the standard deviation of all data.

In this research, NDVI is used as a variable for the *Drought Severity Index* (DSI) according to:

$$\text{DSI} = \frac{\text{NDVI} - \overline{\text{NDVI}}}{\sigma} \tag{3}$$

where NDVI is the mean value of the vegetation index (VI) in each month, $\overline{\text{NDVI}}$ is the average of NDVI for the whole period of study, and $\sigma$ denotes the standard deviation of all long-term data.

According to this, the DSI calculation for any location is based on the long-term NDVI estimated for a desired period. This long-term record is fitted to a probability distribution, which is then transformed into a normal distribution so that the mean DSI for the location and desired period is zero. Positive and negative DSI values indicate values above and below the median NDVI, respectively. Because DSI is normalized, wetter and drier climates can be represented in the same way; thus, wet periods can also be monitored by using DSI [38]. Table 1 shows an example of the hazard quantification procedure.

Frequency is the number of events that an NDVI value will happen within a specific severity category.

Moreover, we have

$$\text{Probability} = \frac{\text{Frequency}}{\text{The number of all years}} \tag{4}$$

$$\text{Percentage} = \text{Probability} \times 100. \tag{5}$$

Weight is a coefficient which increases as severity shifts toward the more negative values. In fact, the weight factor boosts and strenghtens the impact of negative amounts of NDVI as an important indicator of drought. The final relation for total hazard estimation is

$$\text{Total Hazard} = \sum \text{Weight}_{\text{i}} \times \text{Percentage}_{\text{i}}. \tag{6}$$

**Table 1.** Example of hazard quantification procedure.

| Year | NDVI (Mean) | Normal Distributed NDVI | Severity Categories | | | | | |
|---|---|---|---|---|---|---|---|---|
| | | | $(-3)$ to $(-2)$ | $(-2)$ to $(-1)$ | $(-1)$ to $(0)$ | $(0)$ to $(+1)$ | $(+1)$ to $(+2)$ | $(+2)$ to $(+3)$ |
| 1 | X1 | Xn1 | | | | * | | |
| 2 | X2 | Xn2 | | | | | | * |
| 3 | X3 | Xn3 | | | | * | | |
| 4 | X4 | Xn4 | | | * | | | |
| 5 | X5 | Xn5 | * | | | | | |
| 6 | X6 | Xn6 | | | * | | | |
| 7 | X7 | Xn7 | | | | | * | |
| 8 | X8 | Xn8 | | | | * | | |
| 9 | X9 | Xn9 | | | | | * | |
| 10 | X10 | Xn10 | | | * | | | |
| Frequency | | | 1 | 1 | 2 | 3 | 2 | 1 |
| Probability | | | 0.1 | 0.1 | 0.2 | 0.3 | 0.2 | 0.1 |
| Percentage | | | 10 | 10 | 20 | 30 | 20 | 10 |
| Weight | | | 6 | 5 | 4 | 3 | 2 | 1 |

The star symbol (*) indicates the occurring of a drought event with a severity within the range of the severity category.

The second column of Table 1 gives the NDVI average for each month in each year. The normalization of NDVI values (column 3) sets the average and the standard deviation of data to 0 and 1, respectively. Then the range of normalized data will be divided into 6 intervals with assigned weights (1 to 6) based on the intensity of the drought condition.

- Vulnerability assessment (VA):

Vulnerability is the most significant and complex concept in disaster risk analysis. It has many dimensions (e.g., economic, social, demographic, political/institutional, and psychological) that affect people's susceptibility to environmental hazards, in addition to their physical exposure to the hazards themselves, it is influenced by a number of factors, at different levels, from local to global. It also identifies groups that are vulnerable and the factors that make them vulnerable. Furthermore, vulnerability considers the capacities, resources, and assets which people use to resist, cope with, and recover from disasters and other external shocks [39].

The *United Nations Office for Disaster Risk Reduction* (UNDRR) defines vulnerability as *"the characteristics and circumstances of a community, system or asset that make it susceptible to the damaging effects of a hazard"* [40]. Vulnerability is also related to *coping capacity*, which means *"the ability of people, organizations and systems, using available skills and resources, to face and manage adverse conditions, emergencies or disasters"* [41]. It shows the potential of damages from hazard in a certain place. Numerous studies have been conducted applying the vulnerability concept in drought.

Considering that the rate of susceptibility to the impacts of a disaster is not equal for all elements at risk, their degree of vulnerability is also different [41]. For this reason, during the process of vulnerability assessment we often need to define coefficients which are usually called weights (scores) in order to estimate the rate of susceptibility in each element (class) to a disaster. The weighting process needs an expert survey to be conducted based on previous experiences involved in disaster vulnerability assessment. In each survey, experts are asked to conduct a weighting process according to the susceptibility of each element to drought impacts [42]. There is also another type of weighting technique based on vulnerability index [43] which uses the *Standardized Drought Vulnerability Index* (SDVI).

As an example, we can compare farmlands to rangelands, in terms of vulnerability to drought. Farmlands, due to higher water demand, are more vulnerable to drought than rangelands. As a result, the weight assigned to farmlands must be larger than that

assigned to rangelands. The experts working for different sectors relevant to drought management are being asked to assign a weight for each element based on their experiences and knowledge. The final average of the recommended weights by all the experts which are also called "expert opinions" will be assigned to each element (class) [42].

- Identification of exposure or elements at risk (IER):

Exposure is one of the most important concepts in disaster risk analysis. Based on the exact definition of UNDRR terminology, exposure to some natural hazards may be described as "being in the wrong place at the wrong time" [43]. In the case of drought, exposure usually focuses on life damages and losses [43], which is determined by several factors, such as population and livestock density, utilization of land for agriculture (percentage of irrigated farms), as well as water extraction, especially for the industrial sector [44]. Exposure mapping sometimes implies the estimation of population and the number of infrastructures which are under the impact of disasters consequences.

Finally, drought risk is computed as [45]

$$\text{Risk} = \text{Hazard} \times \text{Vulnerability} \times \text{Exposure.} \tag{7}$$

The three main components of a disaster risk assessment have been described in the preceding paragraphs. However, a review of the literature will show that researchers have mostly used one or two components, at most, in their risk-assessment procedure, and rarely quantify all three components. For instance, in one group of drought-risk evaluation papers, researchers only implement hazard quantification. They often use meteorological indices such as the *Standard Precipitation Index* (SPI), the *Standard Precipitation Evapotranspiration Index* (SPEI) and the *Palmer Drought Severity Index* (PDSI) based on ground-collected data [46]. Sometimes they also apply indices provided by remote-sensing data to quantify drought hazard severity [15,47]. Moreover, they might use both remote-sensing and meteorological indices [9,48].

In some papers, methodologies focus on both hazard and vulnerability. For example, researchers in one paper determined drought risk in Iran by using layers such as SPI for hazard quantification, drainage density, and climate data for vulnerability assessment by using ground-based and statistical data [49]. Other researchers produced risk maps by using hazard quantification with the help of SPEI and vulnerability assessment by using statistical data and various other factors such as unemployment and illiteracy rate, the ratio of women to men, and the ratio of the rural to the urban population for the Kerman province located in the southern part of Iran [50].

In a group of vulnerability-based approaches, some researchers have used *Protection Motivation Theory* (PMT) to measure farmers' drought risk management behavior [50–56]. They used a method based on the current behavior of farmers related to the agricultural drought risk management in different parts of Iran. For this purpose, they conducted a survey in order to measure variables (e.g., perceived vulnerability, perceived severity, self-efficacy, response cost, response efficacy and intention) indicating drought vulnerability rate.

As has already been mentioned, this work uses all three abovementioned components to produce a drought risk map in Iran.

### 2.4. Research Data and Period

In this research, the long-term, low-resolution daily NOAA/AVHRR NDVI product, with spatial resolution of 1 km, has been used to prepare monthly composite vegetation indices of May at the national scale from 1986 to 2016. All necessary data required by the study are available through the NOAA website https://www.ncei.noaa.gov (accessed on 27 August 2020).

The first *Advance Very High-Resolution Radiometer* (AVHRR) sensor was launched on board TIROS-N in 1978. This sensor was installed on NOAA-6 in 1979. The third generation of AVHRR (AVHRR/3) on board NOAA-16 was launched in 2010, has 6 channels, and

receives electromagnetic waves in the visible and near-infrared range. Table 2 shows information for the 6 bands of the AVHRR/3 instrument. The last NOAA satellite series is NOAA 19, which was launched in 2019 [20].

**Table 2.** Information of the bands of the AVHRR/3 instrument.

| Channel | AVHRR/3 |
|---------|---------|
| 1 | 0.58–0.68 μm (VIS) |
| 2 | 0.725–1.1 μm (NIR) |
| 3A | 1.58–1.64 μm (NIR) |
| 3B | 3.55–3.93 μm (MIR) |
| 4 | 10.30–11.30 μm (TIR) |
| 5 | 11.5–12.5 μm (TIR) |

As already mentioned, NDVI is defined as the ratio of surface reflectance (SR) differences between NIR and the red band to their sum (relation 1). Based on the Table 2, for AVHRR

$$\text{NDVI} = (\text{Channel2} - \text{Channel1})/(\text{Channel2} + \text{Channel1}). \tag{8}$$

The schematic diagram of the methodology of this research is shown in Figure 5.

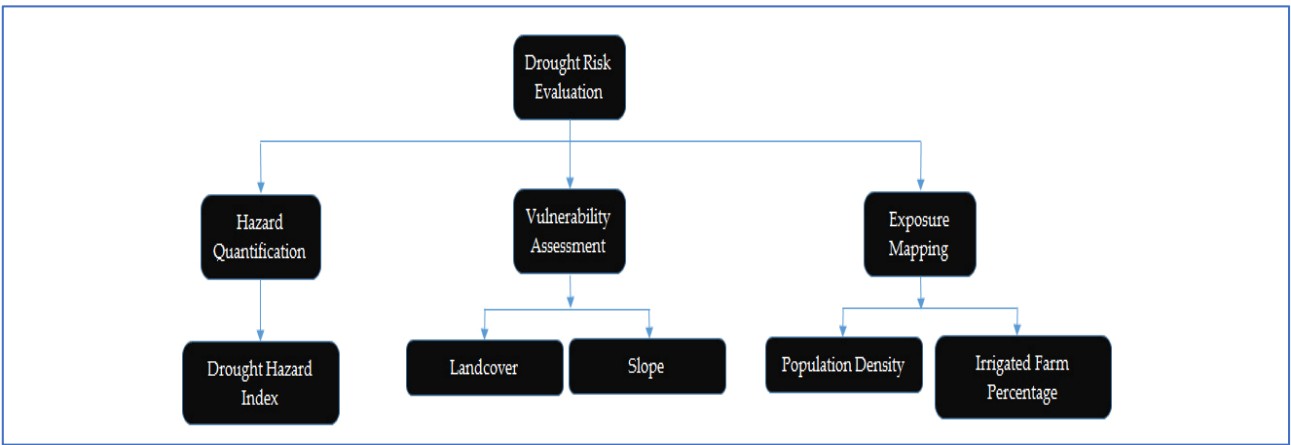

**Figure 5.** Block diagram of the research steps.

## 3. Results

### 3.1. Hazard Quantification

As a first step, the probability of drought occurrence can be stated by determining the vegetation changes during 31 years (1986–2016) by using NDVI. Figure 6 shows the output image of the Global NOAA NDVI product for May 2014 and NDVI of the current study area (Iran) clipped from the global image.

In order to carry out a statistical analysis, monthly NDVI averages during the period of study (1986 to 2016) were calculated for each province in Iran. As an example, the result of the hazard calculation procedure for the Sistan & Baloochestan province (located in the southeast of Iran) is shown in Table 3.

Table 4 gives the results of the hazard calculations for all provinces in Iran, and Figure 7 shows the final hazard map of Iran by using DSI. Based on the map, drought hazard quantities have been categorized between 1 and 0 indicating *The Most Hazardous* and *The Least Hazardous* areas, respectively.

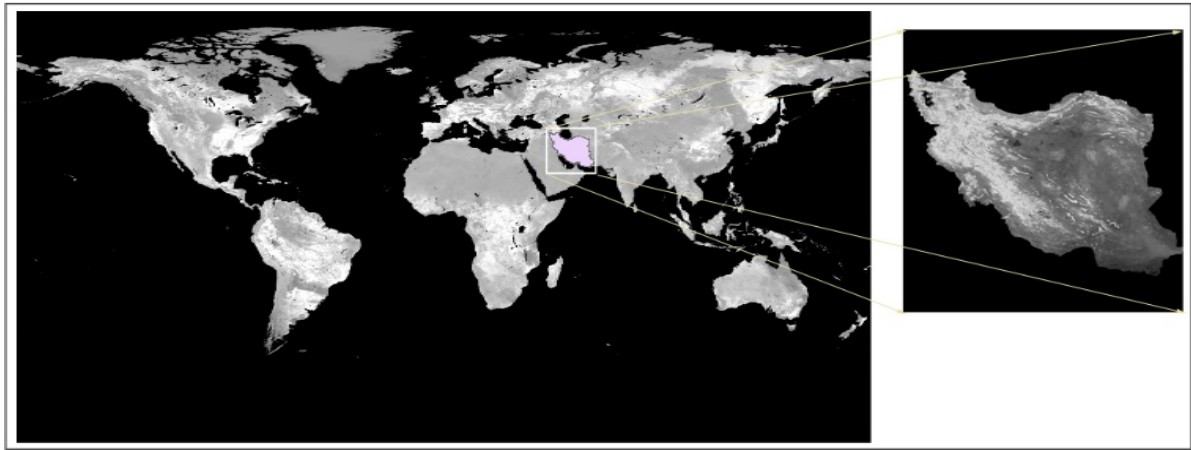

**Figure 6.** Monthly global NOAA NDVI for May 2014 and area of study (Iran). Downloaded product through: https://earthexplorer.usgs.gov/ (accessed on 19 July 2020).

**Table 3.** Hazard quantification for the Sistan & Baloochestan Province (1986–2016).

| Year | NDVI (Mean) | Normal NDVI | Severity Categories | | | | | |
|---|---|---|---|---|---|---|---|---|
| | | | (−3) to (−2) | (−2) to (−1) | (−1) to (0) | (0) to (+1) | (+1) to (+2) | (+2) to (+3) |
| 1986 | 0.19 | 0.59 | | | | * | | |
| 1987 | 0.17 | 0.50 | | | | * | | |
| 1988 | 0.22 | 0.74 | | | | * | | |
| 1989 | 0.28 | 1.03 | | | | | * | |
| 1990 | 0.27 | 0.98 | | | | | * | |
| 1991 | 0.24 | 0.84 | | | | | * | |
| 1992 | 0.17 | 0.50 | | | | * | | |
| 1993 | 0.19 | 0.59 | | | | * | | |
| 1994 | 0.14 | 0.35 | | | | * | | |
| 1995 | 0.12 | 0.25 | | | | * | | |
| 1996 | 0.02 | −0.21 | | | | * | | |
| 1997 | 0.48 | 2.02 | | | | | | * |
| 1998 | 0.03 | −0.18 | | | | * | | |
| 1999 | −0.21 | −1.36 | | | * | | | |
| 2000 | 0.07 | 0.02 | | | | * | | |
| 2001 | 0.38 | 1.50 | | | | | * | |
| 2002 | 0.27 | 1.00 | | | | | * | |
| 2003 | 0.29 | 1.08 | | | | | * | |
| 2004 | −0.17 | −1.16 | | | * | | | |
| 2005 | −0.11 | −0.85 | | | * | | | |
| 2006 | −0.19 | −1.24 | | | * | | | |
| 2007 | −0.30 | −1.79 | | * | | | | |
| 2008 | −0.22 | −1.40 | | * | | | | |
| 2009 | −0.23 | −1.46 | | * | | | | |
| 2010 | −0.06 | −0.64 | | | * | | | |
| 2011 | 0.13 | 0.31 | | | | * | | |
| 2012 | −0.18 | −1.23 | | | * | | | |
| 2013 | 0.08 | 0.07 | | | | * | | |
| 2014 | 0.12 | 0.25 | | | | * | | |
| 2015 | 0.13 | 0.29 | | | | * | | |
| 2016 | −0.21 | −1.37 | | | * | | | |

**Table 3.** *Cont.*

| Year | NDVI (Mean) | Severity Categories | | | | | | |
|---|---|---|---|---|---|---|---|---|
| | | Normal NDVI | (−3) to (−2) | (−2) to (−1) | (−1) to (0) | (0) to (+1) | (+1) to (+2) | (+2) to (+3) |
| Average | 0.07 | | | | | | | |
| Standard Deviation | 0.21 | | | | | | | |
| Frequency | | 0 | 3 | 7 | | 14 | 6 | 1 |
| Probability | | 0 | 0.10 | 0.23 | | 0.45 | 0.19 | 0.03 |
| Percentage | | 0 | 10 | 23 | | 45 | 19 | 3 |
| Weight | | 6 | 5 | 4 | | 3 | 2 | 1 |
| Hazard | | 0 | 50 | 92 | | 135 | 38 | 3 |
| Total Hazard | | | | | | | | 318 |

The star symbol (*) indicates the occurring of a drought event with a severity within the range of the severity category.

**Table 4.** Results of hazard calculations in each Iranian province.

| Provinces | Total Hazards | Provinces | Total Hazards |
|---|---|---|---|
| Hormozgan | 361.9 | Kordestan | 347.62 |
| Hamedan | 361.9 | Kohgilooyeh | 347.62 |
| Tehran | 357.14 | Kerman | 347.62 |
| Khorasan Razavi | 357.14 | Golestan | 347.62 |
| Isfahan | 357.14 | Khuzestan | 347.62 |
| Semnan | 357.14 | Kermanshah | 347.62 |
| Lorestan | 325.38 | Yazd | 346 |
| Southern Khorasan | 325.38 | Ardebil | 346 |
| Ilam | 325.38 | Zanjan | 342.86 |
| Gilan | 325.38 | Northern Khorasan | 342.86 |
| Fars | 325.38 | East Azarbaijan | 342.86 |
| West Azarbaijan | 325.38 | Sistan & Baloochestan | 318 |
| Qazvin | 325.38 | Markazi | 328.57 |
| Shahrekord | 347.62 | Booshehr | 276 |
| Qom | 347.62 | Mazandaran | 90.48 |

*3.2. Vulnerability Assessment (VA)*

As a second step, a vulnerability map was prepared by using different data layers such as land cover, topography, climate factors, etc.

The aim of this phase is to evaluate the spatial vulnerability of Iran to drought. To achieve this, the vulnerability map of the region was obtained by analyzing and overlaying the spatial maps of two important parameters in the drought risk context, namely land cover and topography (see the block diagram of Figure 5). For each of the above parameters, a map was created by using GIS, thus showing the range of vulnerability of that parameter in different regions. Finally, both maps have been overlaid and a unique drought vulnerability map has been developed. In the following sections, each parameter is described in detail.

3.2.1. Land Use Map

A land cover map of Iran prepared by the *Remote Sensing Department of the Iranian Space Agency* (ISA) for 2016 was used in this study (Figure 8). This map was produced by using Terra/MODIS data, and it includes 14 classes (i.e., dense Forest, sparse forest, rangelands, farmlands, urban areas, water bodies, lakes, wetlands, salt lakes, clayspan, rock outcrop, plains, clutes, and sands).

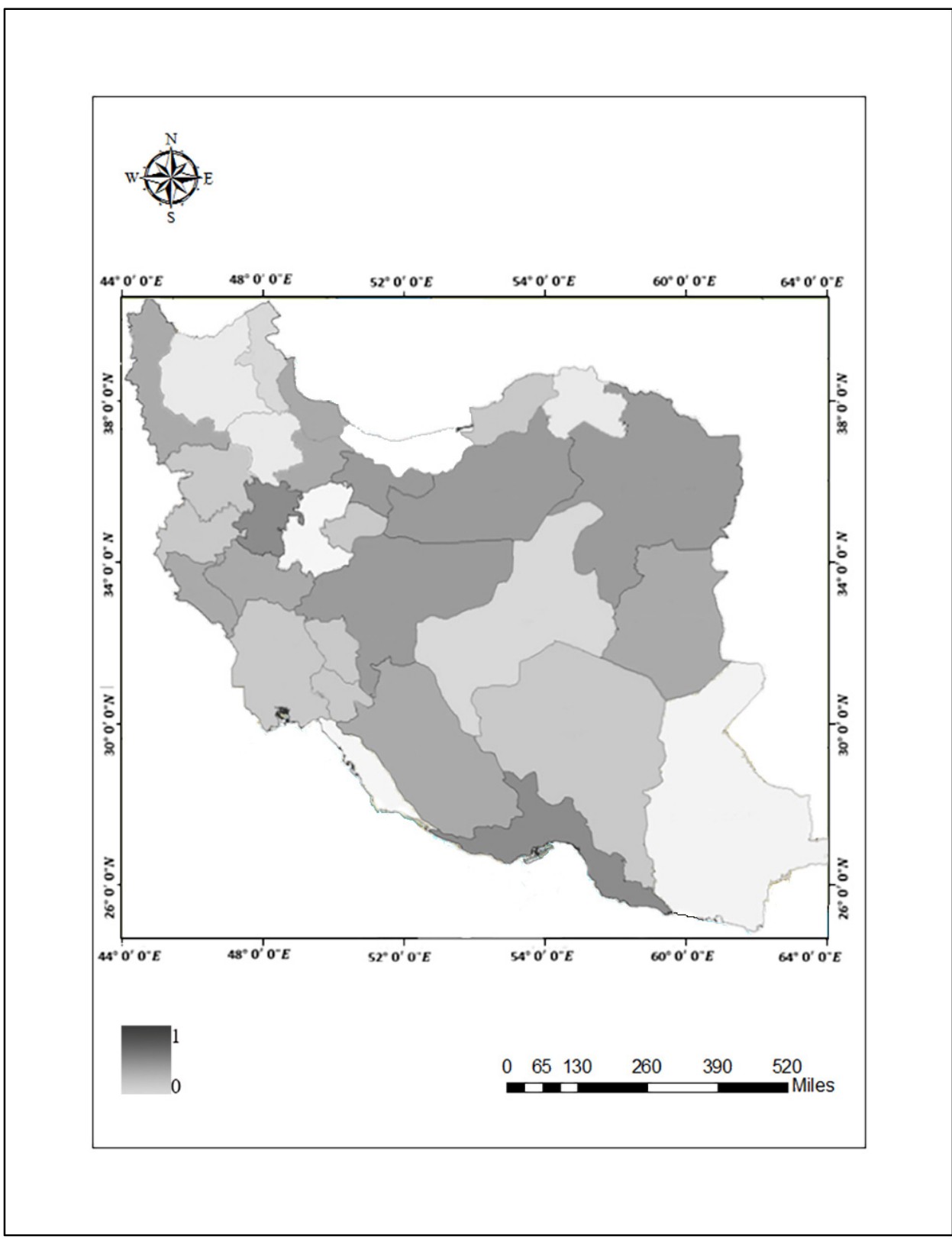

**Figure 7.** Hazard map of Iran using the *Drought Severity Index* calculated by using NOAA/AVHRR satellite data (1986–2016).

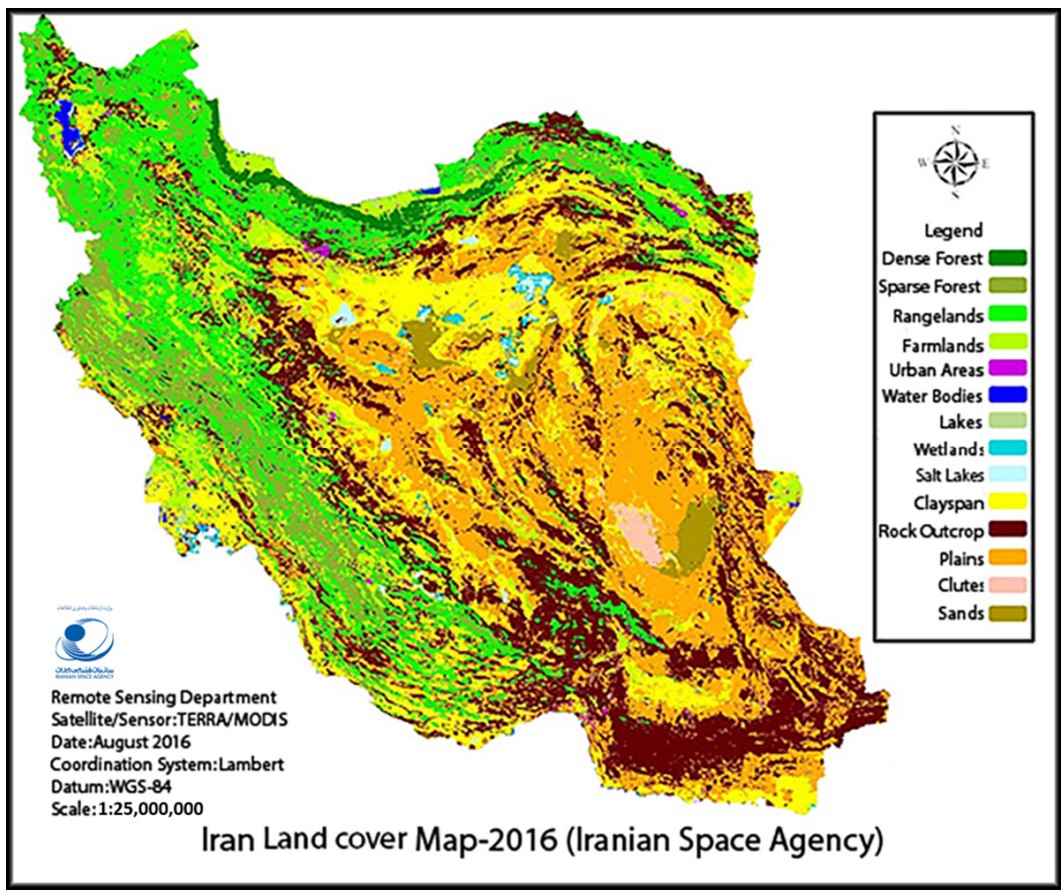

**Figure 8.** Iran land cover pap produced by the *Iranian Space Agency* (2016).

As was already mentioned, in order to prepare a vulnerability map, the opinion of experts to assign weights (scores) for each class is required. In this research, the opinions of 12 experts from three sources in Iran (*Ministry of Agriculture, Ministry of Power*, and *Soil and Water Research Institute*) were collected through a survey conducted in Iran. Table 5 shows the final average scores for each class of the Iran classification map. Each score is categorized from 1 (*Least Vulnerable*) to 5 (*Most Vulnerable*) based on the average of all scores assigned by the experts.

**Table 5.** Land use classes experts' scores for evaluating drought vulnerability.

| Classes | Weights (Scores) |
| --- | --- |
| Farmlands | 5 |
| Rangeland | 4 |
| Sparse Forest | 3 |
| Dense Forest | 2 |
| Salt Lakes | 1 |
| Sands | 1 |
| Clutes | 1 |
| Plains | 1 |
| Rock Outcrop | 1 |
| Clayspan | 1 |
| Wetlands | 1 |
| Lakes | 1 |
| Water Bodies | 1 |
| Urban Areas | 1 |

In the scoring procedure for drought vulnerability regarding the land cover factor, the water demand for each land use type was assumed as a factor that directly influences drought vulnerability. For example, on the one hand, salt lakes, plains, and wetlands have low water demand and, as a result, they are considered to have the lowest vulnerability to drought. On the other hand, irrigated crop areas have a large water demand which makes them the most vulnerable class during water shortage circumstances. The output map after assigning scores to each class is shown in Figure 9.

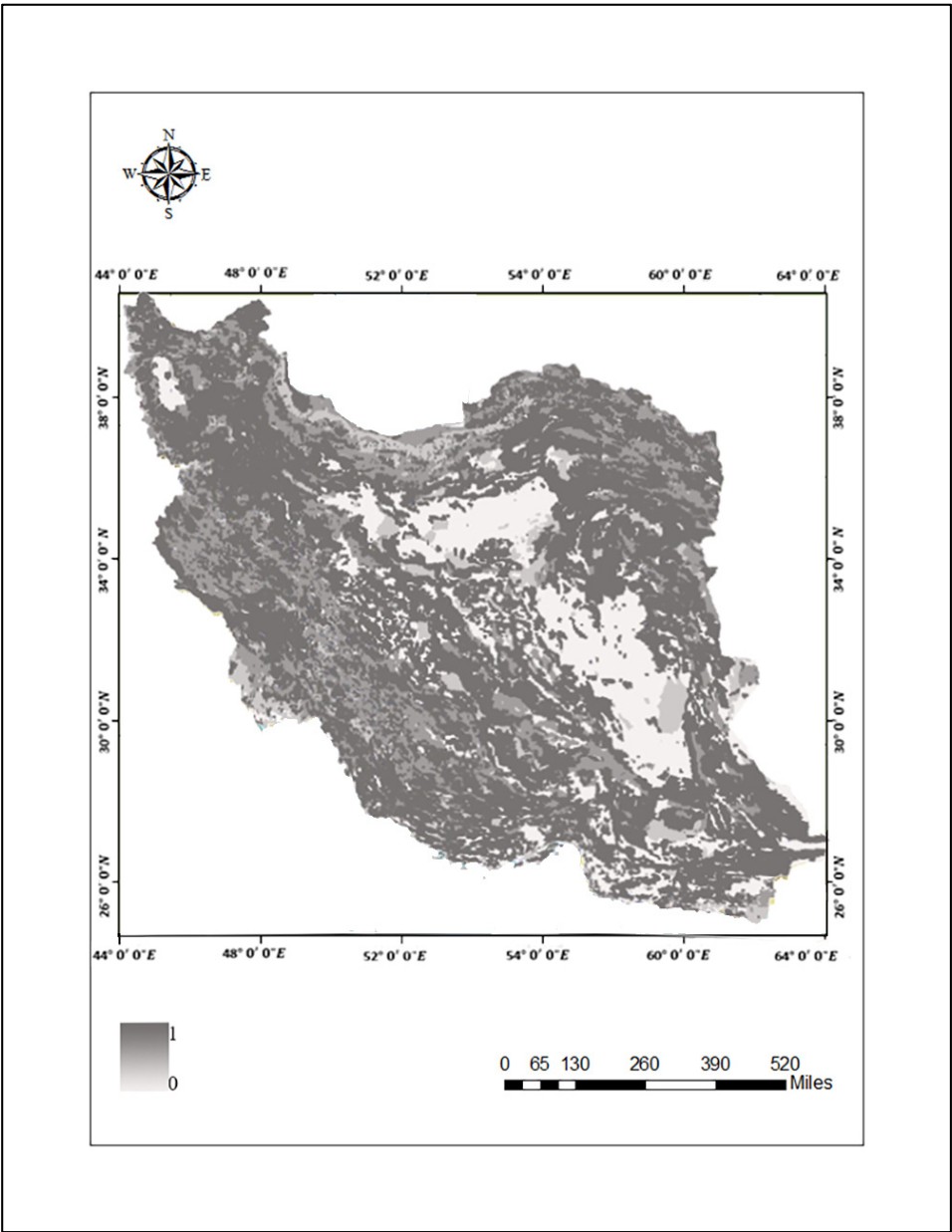

**Figure 9.** Vulnerability map based on landcover (2016).

### 3.2.2. Slope Map

Slope maps represent the topography of the region. Larger slopes may produce larger amounts of runoff, and therefore, less ground water storage could be produced. As a result, the larger slope is considered as more vulnerable to drought. To produce slope maps, 1 arc-second (30 m) *Shuttle Radar Topographic Mission* (SRTM) *Digital Elevation Model* (DEM) is used for Iran, downloaded through the *Earth Explorer* website. Figure 10 then shows the final vulnerability map of Iran resulted from slope and landcover.

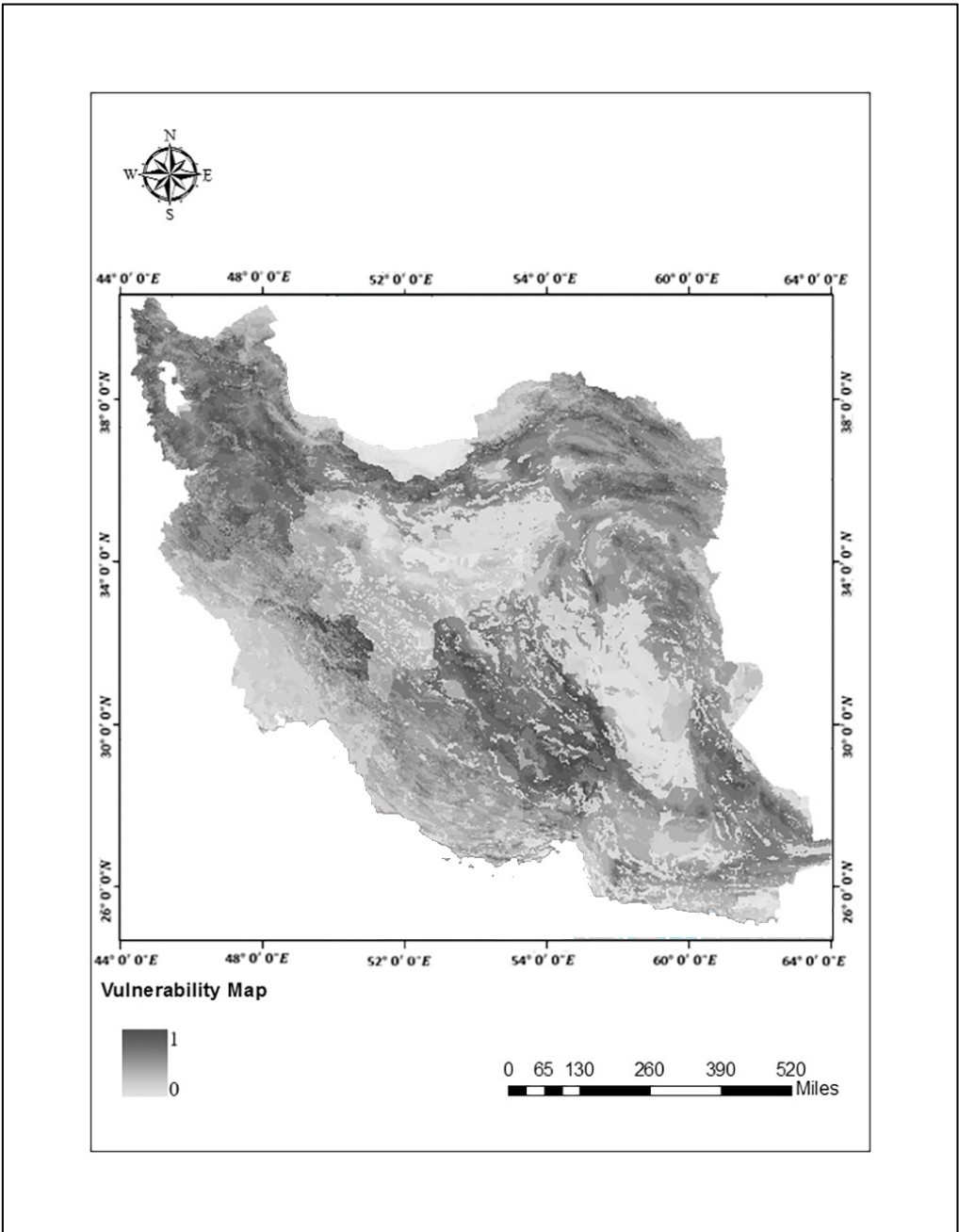

**Figure 10.** Final vulnerability map of Iran (2016).

### 3.3. Exposure Mapping

In this work, two parameters were used in order to prepare the exposure map: (1) population density, and (2) the percentage of irrigated farms. For each parameter, a category map has been created in a GIS environment showing the variability of that parameter in the region. Finally, both category maps have been overlaid, and a unique drought exposure map has been developed.

### 3.3.1. Population Density Map

The population density statistics for each province in Iran (Table 6) was obtained from the *Iranian Census Center* for 2016.

**Table 6.** Population density in Iran for each province (produced by the *Iran Census Center* for 2016).

| Provinces | Population Density (People/km²) | Provinces | Population Density (People/km²) |
|---|---|---|---|
| Tehran | 929.2 | Kordestan | 55 |
| Alborz | 529.4 | Razavi Khorasan | 54.1 |
| Gilan | 180.2 | Bushehr | 51.2 |
| Mazandaran | 137.7 | Markazi | 49.1 |
| Qom | 112.1 | Zanjan | 48.6 |
| Golestan | 91.8 | Isfahan | 47.9 |
| Hamedan | 89.7 | Kohgiluyeh | 46 |
| West Azarbaijan | 87.3 | Fars | 40.2 |
| East Azarbaijan | 85.6 | North Khorasan | 30.4 |
| Qazvin | 81.8 | Ilam | 28.8 |
| Kermanshah | 78.1 | Hormozgan | 25.1 |
| Khuzestan | 73.5 | Kerman | 17.3 |
| Ardebil | 71.4 | Sistan & Baloochestan | 15.3 |
| Lorestan | 62.2 | Yazd | 8.8 |
| Chaharmahal | 58 | South Khorasan | 8.1 |

Figure 11a shows then the Iran exposure map based on population density.

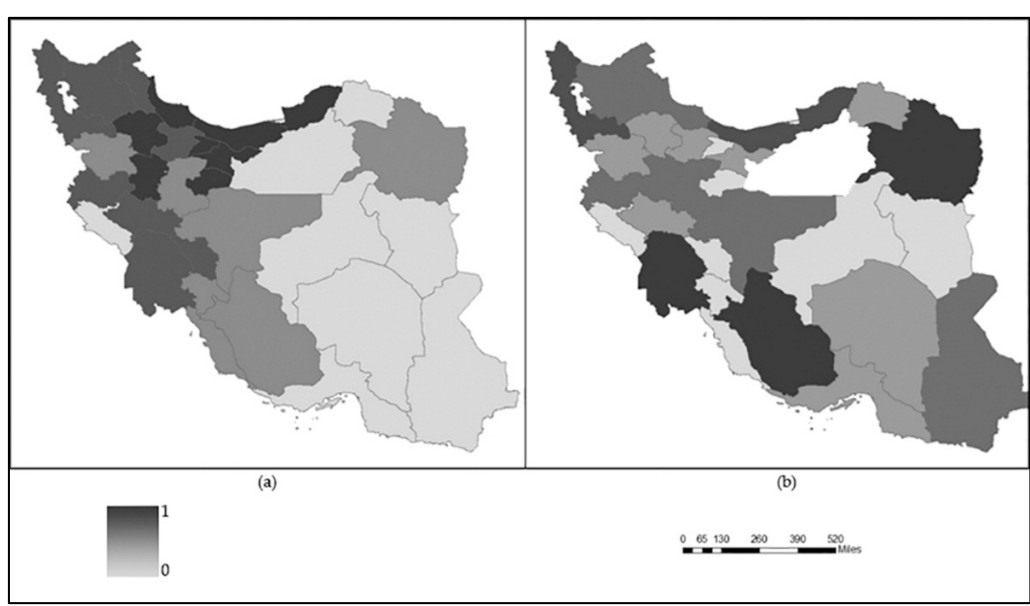

**Figure 11.** (**a**) Exposure map based on population density (*Iran Census Center*, 2016). (**b**) Exposure map based on irrigation percentage (*Ministry of Agriculture*, 2016).

### 3.3.2. Map of Irrigated Farms Percentage

In this step, the statistics of irrigated farm percentages in different provinces in Iran were collected (*Annual Agricultural Reports, Ministry of Agriculture*, 2016). The results are shown in Table 7. Afterward, the irrigation percentage map of Iran was prepared by using this statistic above mentioned statistics as shown in Figure 11b.

The final exposure map was produced by overlaying the irrigated percentage and the population density layers (Figure 12). The lightest (0) and darkest (1) colors indicate the least and most exposure to drought, respectively.

### 3.4. Risk Map

The drought risk map was prepared (Figure 13) by overlaying the hazard, vulnerability, and exposure maps by using the relation (4). It is also essential to know which provinces are at risk of drought in order to prepare a national drought plan (Figure 14).

**Table 7.** Percentage of irrigated farms in Iranian provinces (*Ministry of Agriculture*–2016).

| Provinces | Irrigated Farms (%) | Provinces | Irrigated Farms (%) |
|---|---|---|---|
| Khuzestan | 13.1 | Qazvin | 2.6 |
| Fars | 9 | Lorestan | 2.5 |
| Razavi Khorasan | 8.6 | Kerman | 2.3 |
| South Khorasan | 8.6 | Tehran | 2.2 |
| Golestan | 5.9 | North Khorasan | 1.8 |
| Mazandaran | 5 | Zanjan | 1.8 |
| West Azarbaijan | 5 | Kordestan | 1.5 |
| East Azarbaijan | 4.1 | Hormozgan | 1.3 |
| Hamedan | 3.9 | Chaharmahal | 1.2 |
| Ardebil | 3.6 | Semnan | 1.2 |
| Isfahan | 3.5 | Ilam | 1.1 |
| Gilan | 3.3 | Booshehr | 0.8 |
| Sistan & Baloochestan | 3.1 | Kohgiluyeh | 0.7 |
| Kermanshah | 2.8 | Qom | 0.7 |
| Markazi | 2.7 | Yazd | 0.6 |

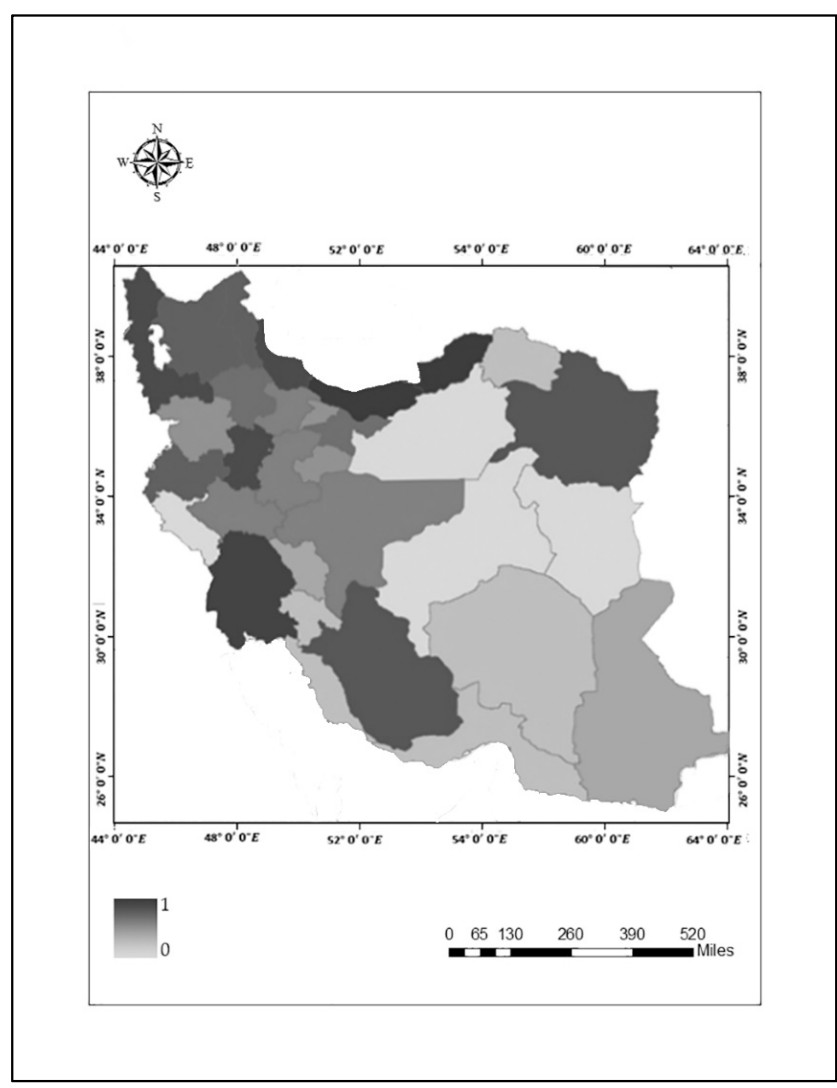

**Figure 12.** Final exposure map of Iran (2016).

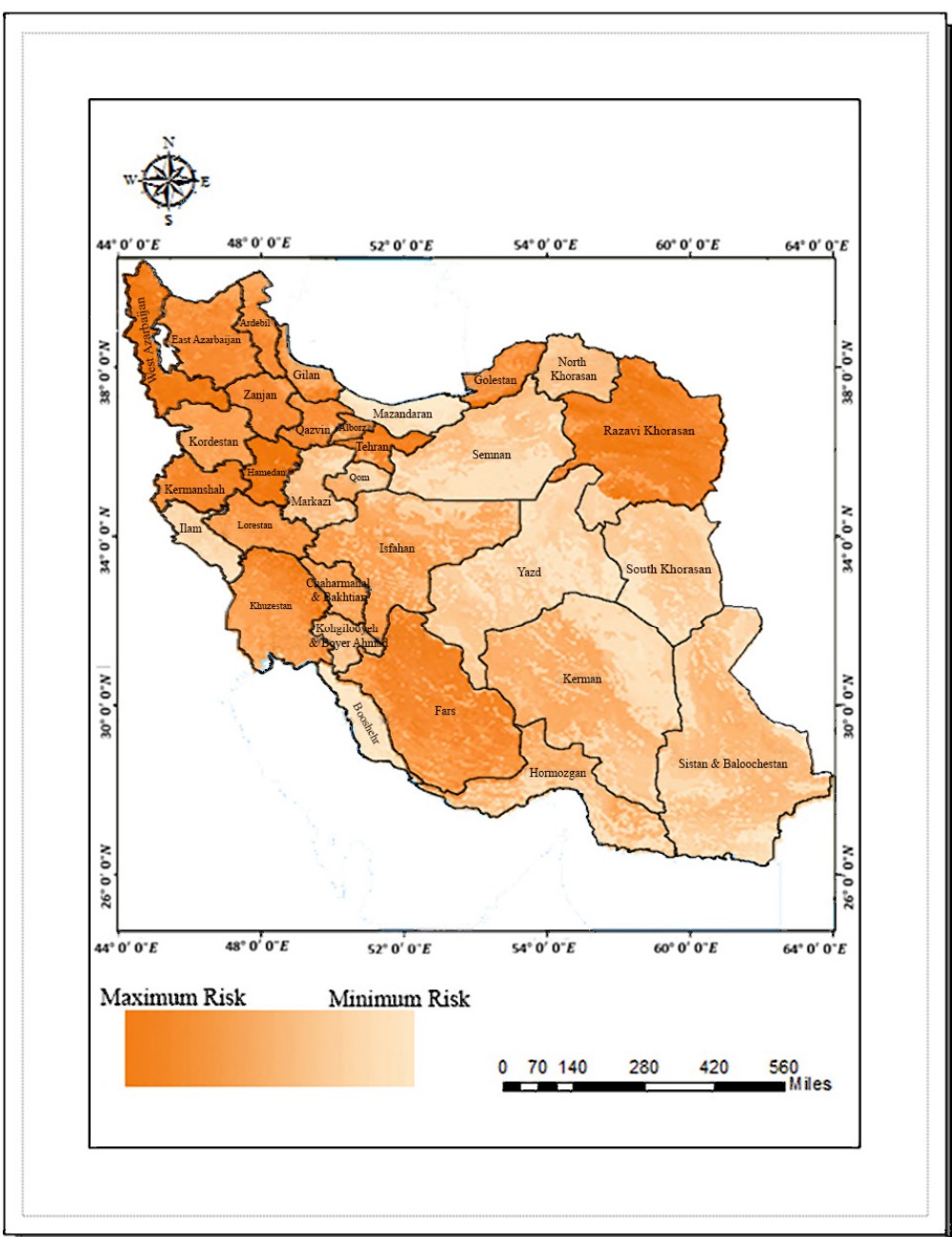

**Figure 13.** Iran drought risk map (2016).

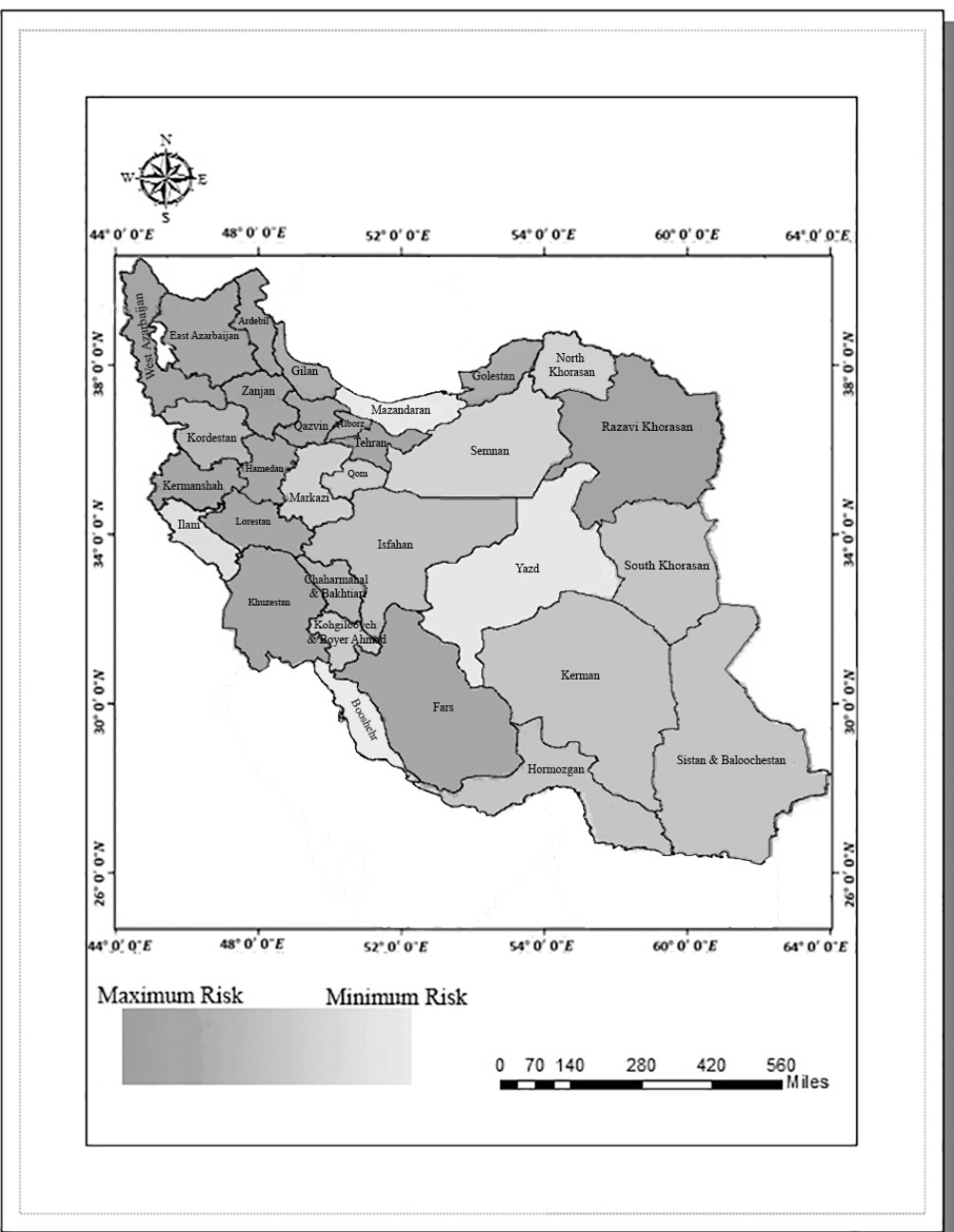

**Figure 14.** Iran drought risk map at a provincial level (2016).

## 4. Discussion

Based on the results obtained, the middle part of Iran, with desert and semi-desert terrain, experiences the least drought risk. The lack of factors such as population, agricultural farms, and any other types of vegetation classes like forests, rangelands etc., moves the overall risk value to near zero in this area. Naturally, it should be mentioned that a negligible value in risk may encourage decision makers to conclude that "*there is no need to plan for any drought mitigation programs in the region*" but, in fact, some forms of activities should be implemented in this area in order to combat desertification by integrating land and water management. The activities to protect soil from erosion, salinization, and other types of degradation are examples which are recommended to avoid desertification.

In contrast, the northwest of Iran, with a climate that corresponds to central European weather conditions (cold, wet and a few enjoyable summer months), suffers from a high drought risk value. This area enjoys vegetated and elevated areas, often with a good

rate of precipitation compared to the other parts of the country. One major reason for high-risk values in this region is the population density, which shows a strong amount of exposure to drought conditions. The high percentage of agricultural irrigated farms, as another exposure factor, is the second reason for increasing the drought risk value in the above region.

Another area in the northeast of the country, the Razavi Khorasan province, is also at high risk of drought, especially compared to the other surroundings provinces. The large amount of risk value in this area is also due to two above exposure factors, but the difference is that the role of irrigated farm percentage is much stronger than population density in this region. As seen from Table 7, Razavi Khorasan is the third province from the aspect of irrigated farms percentage (8.6%), but the seventeenth-ranked province in terms of population density.

Moreover, based on the risk map, the southern part of the Caspian Sea shows a very low drought risk. The Mazandaran province is located in this region, where low drought frequency during the period of study is an important factor in decreasing the risk value (Table 3). Mazandaran is isolated between the Caspian Sea in the north and the Alborz mountains in the south, which will be influenced by the humid weather (caused by sea) and cold weather (coming from the mountains). For this reason, the climate in this province is mostly moderate and subtropical with an average temperature of 25 °C and 8 °C in summer and winter, respectively [57], which finally decreases significantly the total drought hazard and risk (90.48).

The Booshehr province in the south of Iran, with a hot, semi-arid climate shows, on the one hand, a low amount of drought, which is totally different from the other provinces located around it. On the other hand, it is one of the warmest regions in Iran with an average daily high temperature of 30 ºC. However, its lowest drought risk value based on the results of this research, is due to the low percentage of irrigated farms (0.8) and population density (51.2), as well as the low drought hazard (DHI = 276).

## 5. Conclusions

In recent decades, the risk phase of disaster management has attracted more attention among managers than traditional crisis management. Risk management consists of two phases called "risk analysis" and "risk assessment", and this paper focuses on the latter by quantifying drought risk in Iran on a national scale. For this purpose, five information layers, including the drought severity index, landcover and slope maps, population density, and irrigated farm percentage have been prepared and compounded by using remote sensing and GIS techniques. The results show that the main areas of the country, in the west and northwest, and a small part in the northeast of Iran are highly threatened by the high risk of drought. In conclusion, the output drought risk map in this research shows that decision makers should allocate more money in their budget to the provinces located in the west, northwest, and small parts of the northeast of Iran (with valuable factors such as millions of inhabitants, vast agricultural and vegetated areas) than the areas located in the deserts (central parts of Iran) without any population, vegetation, or human elements.

The results of drought risk evaluation (provided by remote sensing and GIS experts) should be assimilated with the outputs of drought risk analysis—prepared by the experts from different disciplines such as hydrology, meteorology, agriculture, etc.—to develop a list containing advice to help high-level decision makers reduce drought risk, mitigate drought impacts, develop future drought scenarios, and follow an adaptation strategy. In future work, the addition of other significant data provided by remote sensing and GIS techniques, such as precipitation, temperature, soil moisture, drainage density, and water table, will allow researchers to assess and map drought risk more accurately. Furthermore, as some Studies Have Shown, the *Enhanced Vegetation Index* (EVI) is more sensitive to drought stressors [58] and may provide more precise drought risk evaluation maps and is therefore recommended for future work. Regarding the vulnerability weighting method, using

indices such as the *Standardized Drought Vulnerability Index* (SDVI) will be recommended as an effective method for vulnerability assessment in future research.

**Author Contributions:** Conceptualization, A.A.A., H.A. and E.L.-B.; Investigation, A.A.A.; Methodology, A.A.A.; Project administration, A.A.A.; Writing—original draft, A.A.A.; Writing—review & editing, H.A. and E.L.-B. All authors have read and agreed to the published version of the manuscript.

**Funding:** This research received no external funding.

**Conflicts of Interest:** The authors declare no conflict of interest.

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
