# Peer review of "Drought Risk Evaluation in Iran by Using Geospatial Technologies"

_remotesensing, doi:10.3390/rs14133096_

Round 1

Reviewer 1 Report

See the attached file for my comments

Author Response

Response to Reviewer 1 Comments

Many thanks for your very effective and helpful comments on our manuscript. Please see below our responses to your suggestions and requirements:

Comment No. 1: Authors should re-arrange the keywords alphabetically:

Response: The keywords were arranged alphabetically.

Comment No. 2: The purpose of the study can be improved for clarity

Response: The paragraph explaining the purpose of the study has been improved as shown below:

During the recent decades, drought managers have distinguished the essential of drought risk management with the aim of mitigating drought impacts [8]. In most researches about drought risk assessment, the final risk map shows only drought hazard” [9-13], “vulnerability” [14,15],” exposure"[16], or “hazard and vulnerability “in one research [17]. On the other hand, by considering that Earth Observation satellite techniques offer new opportunities to understand drought risks [18], some references which some of them mentioned above [5,7,8,13] have used remote sensing data to analyze drought risk.

The aim of this paper is the evaluation (mapping) of drought risk in Iran by quantifying all three elements of risk (hazard, vulnerability and exposure) using remote sensing NOAA/AVHRR satellite data since 1986 to 2016, land cover map, slope map as well as statistical data such as population density and irrigation farm percentage to provide risk map in the national and provincial scale in a GIS based environment. To the best of the authors knowledge and review, this is the first work about Iran drought risk in which all the three main phases of risk management, i.e. hazard, vulnerability, and exposure, using satellite images and GIS, have been evaluated in one piece of research.

Comment No. 3: Study Area” should come immediately under “Materials & Methods”

Response: “Study Area” has been moved to the first part of the “Materials & Methods”

Comment No. 4: Figures 5 & 6, can be moved to the sub file.

Response: Both figures were merged into one figure.

Comment No. 5: This section should be improved; authors compare their findings with previous studies   see few suggestions below;

https://doi.org/10.1016/j.jenvman.2021.112028

https://doi.org/10.1016/j.jhydrol.2018.11.058

https://doi.org/10.1007/s00477-019-01648-4

Response: Although the targets of the above references are a bit different from our manuscript’s aim, two out of the three above references have been included in the conclusions, for further instructions in the future.

For example, in https://doi.org/10.1016/j.jenvman.2021.112028, the target is drought monitoring which is suitable for “Drought Damage Assessment”. But the target of our manuscript is “Drought Risk Assessment” which is different from “Drought Damage Assessment”. However, a conclusion from the above paper stating that ”EVI index is more sensitive to drought stressors” has been added to the conclusion of our manuscript as a good suggestion for future research and is referred as reference No. 60.

Regarding https://doi.org/10.1007/s00477-019-01648-4 we have added it as reference Number 25 and in the conclusions, we recommend using SDVI in future research for vulnerability scoring.

Reviewer 2 Report

The method used in the paper looks like very artificial (as many similar methods for different risk assessment) and I do not believe that it has practical utility.

Because of this of crucial importance is that authors add one or two chapters in which they should give information about long lasting time series of air temperature and precipitation as crucial factors for drought occurring in past, present and in future.

It will be useful to give more data about real drought in Iran. For example, is it true that the North West of the country in reality are exposed to the maximum drought risk, and in the middle of Iran, exhibits minimum risk against drought. Is it real or only result (conclusion) of this approach (methodology, model?).

It seems to me extremely illogical that minimum risk (see Fig. 17) occurred in hyper arid region (see Fig. 3).

There are too much maps. Some of them are not important.

Author Response

Response to Reviewer 2 Comments

Many thanks for your very effective and helpful comments on our manuscript. Please see below our responses to your suggestions and requirements:

Comment No.1: Because of this of crucial importance is that authors add one or two chapters in which they should give information about long lasting time series of air temperature and precipitation as crucial factors for drought occurring in past, present and in future.

Response:

We appreciate this suggestion very much which we consider as very useful. Then, the sub-section entitled “2.2. Long Term Air temperature and Precipitation Trend in Iran” has been added to the “Materials and Methods” section as complementary information.

 Comment No.2:

It will be useful to give more data about real drought in Iran. For example, is it true that the North West of the country in reality are exposed to the maximum drought risk, and in the middle of Iran, exhibits minimum risk against drought. Is it real or only result (conclusion) of this approach (methodology, model?). It seems to me extremely illogical that minimum risk (see Fig. 17) occurred in hyper arid region (see Fig. 3).

Response:

Please note that the target of our research is “Drought Risk Assessment” which differs from “Drought Damage Assessment”. In most research about drought risk assessment, researchers calculate just “hazard” as final risk. In other words, they use only the following relation for risk assessment:

Risk = Hazard

In the above case, the final risk map would be “Figure (7)- Hazard Map of Iran using the Drought Severity Index calculated using NOAA/AVHRR satellite data (1986-2016)” where the north west of Iran shows small risk and the middle parts show the biggest amount of risk confirming the reviewer conclusion.

But when we use the complete risk quantification relation (as in our manuscript):

Risk = Hazard x Vulnerability x Exposure

the results and the conclusions will change and the final risk map would be: “Figure (13)- Iran Drought Risk Map (2016)”. This is the result of applying “Vulnerability” (Landuse & Slope) and especially “Exposure” (Irrigated farms & population density) factors in the above relation.

But what is the purpose of using the above relation? Basically, when we prepare a (disaster) risk map, we are planning to protect people and assets from the disaster impacts and consequences in the future. In other words, people and any elements made by human, play more important role in a precise disaster risk assessment. As a result, the regions with more population and more manmade factors such as constructional elements (especially in the cases of earthquake and floods), agricultural (in the case of drought), infrastructures, social or cultural assets (for all types of disasters) are more sensitive to the disaster impacts than the regions which are empty and free of people.

In conclusion, the output drought risk map in this research is showing that decision makers should allocate more budget to the north west of Iran (with valuable conditions such as millions of inhabitants, vast agricultural and vegetated areas) than the areas located in the deserts (middle parts of Iran) without any population, vegetation and human elements.

Finally, even if our target in this paper was “Drought Damage Assessment”, vegetation losses in the north west of Iran with dense cover would be significantly worse than those in the deserts located in the middle parts of Iran without any vegetation. In other words, the vegetated areas in the northwest of Iran suffers from drought damages, more than the desert areas in the middle of Iran without any vegetation.

The following paragraph was added to the manuscript conclusion for more clarification

“In conclusion, the output drought risk map in this research is showing that decision makers should allocate more budget to the provinces located in the west, northwest and small parts of the northeast of Iran (with valuable factors such as millions of inhabitants, vast agricultural and vegetated areas) than the areas located in the deserts (Middle parts of Iran) without any population, vegetation and human elements.

 Comment No.3:

There are too much maps. Some of them are not important.

Response:

Figures 5 & 6 were merged.

Figure 12 was removed.

Figure 16 was removed. 

Round 2

Reviewer 1 Report

Authors have addressed my comments

Reviewer 2 Report

No comment